Estimation of the percentile of Birnbaum-Saunders distribution and its application to PM2.5 in Northern Thailand

Thangjai Warisa 1
http://orcid.org/0000-0001-8269-3397 Niwitpong Sa-Aat 2 sa-aat.n@sci.kmutnb.ac.th
http://orcid.org/0000-0003-3059-1131 Niwitpong Suparat 2
1 Department of Statistics, Ramkhamhaeng University , Bangkok , Thailand
2 Department of Applied Statistics, King Mongkut’s University of Technology North Bangkok , Bangkok , Thailand
Gavrilescu Maria
Electronic publication date: 2024 Feb 29
Publication date: 2024
Volume: 12
Electronic Location ID: e17019
Received 2023 Nov 1; Accepted 2024 Feb 6
Copyright: © 2024 Thangjai et al.
Copyright year: 2024
Copyright holder: Thangjai et al.
License: This is an open access article distributed under the terms of the Creative Commons Attribution License, which permits unrestricted use, distribution, reproduction and adaptation in any medium and for any purpose provided that it is properly attributed. For attribution, the original author(s), title, publication source (PeerJ) and either DOI or URL of the article must be cited.
License URL: https://creativecommons.org/licenses/by/4.0/

Keywords: Bayesian approach, Birnbaum-Saunders distribution, Bootstrap approach, Generalized confidence interval approach, Percentile

Funding: King Mongkut’s University of Technology North Bangkok KMUTNB-67-KNOW-09 This research was funded by the King Mongkut’s University of Technology North Bangkok. Grant No: KMUTNB-67-KNOW-09. The funders had no role in study design, data collection and analysis, decision to publish, or preparation of the manuscript.

==============================
The Birnbaum-Saunders distribution plays a crucial role in statistical analysis, serving as a model for failure time distribution in engineering and the distribution of particulate matter 2.5 (PM2.5) in environmental sciences. When assessing the health risks linked to PM2.5, it is crucial to give significant weight to percentile values, particularly focusing on lower percentiles, as they offer a more precise depiction of exposure levels and potential health hazards for the population. Mean and variance metrics may not fully encapsulate the comprehensive spectrum of risks connected to PM2.5 exposure. Various approaches, including the generalized confidence interval (GCI) approach, the bootstrap approach, the Bayesian approach, and the highest posterior density (HPD) approach, were employed to establish confidence intervals for the percentile of the Birnbaum-Saunders distribution. To assess the performance of these intervals, Monte Carlo simulations were conducted, evaluating them based on coverage probability and average length. The results demonstrate that the GCI approach is a favorable choice for estimating percentile confidence intervals. In conclusion, this article presents the results of the simulation study and showcases the practical application of these findings in the field of environmental sciences.

Introduction

PM2.5, also known as particulate matter 2.5, refers to tiny particles or droplets suspended in the air with a size not exceeding 2.5 micrometers. These minute particles fall under the category of airborne pollutants and can be composed of a variety of materials, including dust, soil, soot, smoke, and liquid droplets (Thangjai & Niwitpong, 2020). Due to their diminutive size, PM2.5 particles can be readily inhaled into the respiratory system, penetrating deeply. The presence of PM2.5 in the atmosphere raises notable environmental and health concerns, given its potential adverse impact on human health. Long-term exposure to PM2.5 is associated with a spectrum of health problems, including respiratory issues, cardiovascular ailments, and even premature mortality. Short-term exposure to elevated PM2.5 levels can exacerbate conditions such as asthma and bronchitis. The monitoring of PM2.5 levels is a routine practice in assessing air quality, and numerous countries have established air quality standards and regulations to limit the concentration of these fine particles in the air to protect public health. Residents in regions with heightened levels of PM2.5 are frequently advised to take precautions, like staying indoors during periods of poor air quality and using air purifiers to reduce their exposure. Thailand has encountered air quality challenges related to PM2.5, especially during the dry season when activities like agricultural and forest burning contribute to increased levels of particulate matter in the air. Cities like Bangkok, Chiang Mai, Mae Hong Son, and Lampang provinces have experienced periods of subpar air quality due to PM2.5 pollution, prompting health advisories and government interventions to mitigate its impact. Thailand has initiated a variety of measures to combat air pollution, including the implementation of regulations, public awareness campaigns, and steps to reduce pollution sources, such as curbing open burning and enforcing vehicle emission standards. Much like many other governments globally, the Thai government has been actively working to enhance air quality and reduce the health risks associated with PM2.5 exposure. Many researchers have studied PM2.5 air pollution, including Broomandi et al. (2021) and Galán-Madruga et al. (2023).

The Birnbaum-Saunders distribution has become well-known in the fields of life testing and engineering. Its initial application was to model the time until failure, a consequence of the growth of a dominant crack under cyclic stress, where failure occurs once it surpasses a specified threshold (Birnbaum & Saunders, 1969a). Additionally, Bhattacharyya & Fries (1982) illustrated that the Birnbaum-Saunders distribution can function as an approximation of the inverse Gaussian distribution. Desmond (1986) also proposed that this distribution can be considered a balanced combination of the inverse Gaussian distribution and its reciprocal. In the context of statistical inference, Birnbaum & Saunders (1969b) introduced maximum likelihood estimation for the distribution’s parameters. Ng, Kundu & Balakrishnan (2003) put forward a modified moment estimation method for these parameters, along with a technique to mitigate bias in both the maximum likelihood estimator and the modified moment estimator. Wu & Wong (2004) presented approximated confidence intervals for the Birnbaum-Saunders distribution using higher-order likelihood asymptotic procedures. Furthermore, Li & Xu (2016) carried out a comparative study involving the fiducial estimator, maximum likelihood estimator, and Bayesian estimator for the distribution’s unknown parameters. Guo et al. (2017) developed approaches for interval estimation and hypothesis testing pertaining to the common mean of multiple Birnbaum-Saunders populations, utilizing a hybrid methodology that combines generalized inference and large sample theory. Jayalath (2021) applied a flexible Gibbs sampler to draw inferences about the two-parameter Birnbaum-Saunders distribution when dealing with right-censored data. Lastly, Puggard, Niwitpong & Niwitpong (2022) introduced novel techniques for estimating confidence intervals for both the variance and the difference in variances of Birnbaum-Saunders distributions, using PM2.5 data. Based on their work, we subsequently developed confidence intervals for the percentiles of Birnbaum-Saunders distributions, also leveraging PM2.5 data.

In the realm of statistical analysis, while mean and variance are frequently utilized measures, in practical scenarios, percentiles assume greater significance than mean or variance. Percentiles are a commonly used statistical concept. They help determine where an observation stands relative to a specified percentage of values below it, and they serve as indicators of both central tendency and variability. Percentiles are a statistical tool employed to elucidate the relative position of a specific data point within a dataset or a probability distribution. They reveal the percentage of data points that reside either below or above a designated value in the dataset. This methodology is often applied to gain insights into data distributions and to gauge where a particular data point stands in relation to others. The 25th percentile, also known as the first quartile, designates the threshold below which 25% of the data points are situated. It serves as the lower demarcation of the initial quarter of the dataset. The 50th percentile, commonly referred to as the median, is the pivotal point that effectively bisects the dataset into two equal halves, with precisely half of the data points positioned below it and half above it. There is a necessity to compare two distributions, and the researcher will need a specific parameter for the purpose of comparison. Often, researchers commonly select the mean as their primary reference point, presuming it to be the most dependable parameter for characterizing the population. Nonetheless, this preference is not universally applicable, and there are situations where the median emerges as a more reliable reference, especially when dealing with a strongly skewed distribution (Price & Bonett, 2002). Meanwhile, the 75th percentile, or the third quartile, signifies the boundary beneath which 75% of the data points are positioned. It denotes the upper threshold of the initial three-quarters of the dataset. Percentiles are invaluable for comprehending the distribution of data, identifying atypical data points, and facilitating comparative analyses. They are widely used in various fields, including education, where they aid in grading and ranking students, healthcare, for evaluating growth and health metrics, and finance, where they are instrumental in scrutinizing investment returns and associated risks. In situations where PM2.5 levels are high and pose a health threat, percentiles offer a more appropriate option than utilizing the mean and variance. Several researchers have delved into the realm of statistical inference pertaining to percentiles and quartiles. These scholars include Marshall & Walsh (1950), Harrell & Davis (1982), Kaigh & Lachenbruch (1982), Albers & Löhnberg (1984), Cox & Jaber (1985), Chang & Tang (1994), Padgett & Tomlinson (2003), Guo & Krishnamoorthy (2005), Huang & Johnson (2006), Navruz & Özdemir (2018), Hasan & Krishnamoorthy (2018), as well as Abdollahnezhad & Jafari (2018).

A multitude of scholars have delved into statistical investigations regarding the Birnbaum-Saunders distribution. Prominent contributors in this domain comprise Birnbaum & Saunders (1969b), Engelhardt, Bain & Wright (1981), Achcar (1993), Lu & Chang (1997), Ng, Kundu & Balakrishnan (2003), Wu & Wong (2004), Leiva et al. (2008), Wang (2012), Niu et al. (2014), Wang, Sun & Park (2016), and Guo et al. (2017). The objective of this article is to present confidence intervals for percentiles within the Birnbaum-Saunders distribution. Four distinct methods, namely the generalized confidence interval (GCI) approach, the bootstrap approach, the Bayesian approach, and the highest posterior density (HPD) approach, are employed to compute interval estimations for percentiles within the population. These methods make use of simulated data to establish the confidence intervals. To enhance their practical utility, a computer program has been created in the R programming language to compute coverage probability and average interval length. The article includes a numerical example to demonstrate the application of this program.

Methods

Let X=(X1,X2,...,Xn) be a random variable of size n drawn from a Birnbaum-Saunders distribution. The probability density function is defined by

(1) f(x;α,β)=12αβ2π((βx)1/2+(βx)3/2)exp⁡(−12α2(xβ+βx−2));x>0,α,β>0,

where α is the shape parameter and β is the scale parameter.

According to Chang & Tang (1994) and Padgett & Tomlinson (2003), the percentile of X is given by

(2) θ=β4(αzp+α2zp2+4)2,

where zp=Φ−1(p) is the standard normal p-th quantile.

Generalized confidence interval approach

Following Puggard, Niwitpong & Niwitpong (2022), the GPQ for β is given by

(3) Rβ={max(β1,β2);T≤0min(β1,β2);T>0,

where β1 and β2 are two solutions for β can be derived by solving Aβ2−2Bβ+C=0, A=(n−1)J2−1nLT2, B=(n−1)IJ−(1−IJ)T2, C=(n−1)I2−1nKT2, I=1n∑i=1nXi, J=1n∑i=1n1Xi, K=∑i=1n(Xi−I)2, L=∑i=1n(1Xi−J)2, and T∼t(n−1).

Suppose that S1=∑i=1nXi, S2=∑i=1n1Xi, and V∼χ2(n). The GPQ for α is given by

(4) Rα=s2(Rβ)2−2nRβ+s1RβV,

where Rβ is defined in Eq. (3) and s1 and s2 are the observed values of S1 and S2, respectively.

The GPQ for θ is obtained by

(5) Rθ=Rβ4(Rαzp+(Rα)2zp2+4)2,

where zp=Φ−1(p) is the standard normal p-th quantile, Rβ is defined in Eq. (3), and Rα is defined in Eq. (4).

Therefore, the 100(1−γ)% two-sided confidence interval for the percentile of Birnbaum-Saunders distribution using the GCI approach is given by

(6) CIθ.GCI=[Lθ.GCI,Uθ.GCI]=[Rθ(γ/2),Rθ(1−γ/2)],

where Rθ(γ/2) and Rθ(1−γ/2) denote the 100(γ/2)-th and 100(1−γ/2)-th percentiles of Rθ, respectively.

Algorithm 1 was used to construct the GCI for the percentile of Birnbaum-Saunders distribution.

Algorithm 1 Step 1: Generate sample from the Birnbaum-Saunders distribution	
Step 2: Compute A, B, C, S1, S2	
Step 3: At the m step	
(a) Simulate T~t(n−1) and compute Rβ using Eq. (3)	
(b) If Rβ<0, resimulate T∼t(n−1)	
(c) Simulate V∼χ2(n) and compute Rα using Eq. (4)	
(d) Compute Rθ using Eq. (5)	
Step 4: Repeat step 3, a total M times and obtain an array of Rθ’s	
Step 5: Compute Lθ.GCI=Rθ(γ/2) and Uθ.GCI=Rθ(1−γ/2)	

Bootstrap approach

Suppose that x=(x1,x2,...,xn) is a random sample drawn from Birnbaum-Saunders distribution with shape parameter α and scale parameter β. Let α^ and β^ be the maximum likelihood estimators of α and β, respectively. Let x∗=(x1∗,x2∗,...,xn∗) be a bootstrap sample drawn from Birnbaum-Saunders distribution with α^ and β^. Hence, α^∗ and β^∗ are acquired by utilizing B bootstrap samples.

Suppose that b(α^,α) and b(β^,β) are the bias estimators of α^ and β^, respectively. According to Puggard, Niwitpong & Niwitpong (2022), the estimators for b(α^,α) and b(β^,β) are obtained by

(7) b^(α^,α)=1B∑k=1Bα^k∗−α^

and

(8) b^(β^,β)=1B∑k=1Bβ^k∗−β^.

Following MacKinnon & Smith (1998), the respective correct estimates for α^∗ and β^∗ are obtained by

(9) αk~=α^k∗−2b^(α^,α)

and

(10) βk~=β^k∗−2b^(β^,β),

where k=1,2,...,B.

Therefore, the bootstrap estimator of the percentile is obtained as

(11) θ^k=βk~4(αk~zp+(αk~)2zp2+4)2,

where zp=Φ−1(p) is the standard normal p-th quantile, αk~ is defined in Eq. (9), and βk~ is defined in Eq. (10).

Therefore, the 100(1−γ)% two-sided confidence interval for the percentile of Birnbaum-Saunders distribution using the bootstrap approach is given by

(12) CIθ.B=[Lθ.B,Uθ.B]=[θ^k(γ/2),θ^k(1−γ/2)],

where θ^k(γ/2) and θ^k(1−γ/2) denote the 100(γ/2)-th and 100(1−γ/2)-th percentiles of θ^k, respectively.

Algorithm 2 was used to construct the bootstrap confidence interval for the percentile of Birnbaum-Saunders distribution.

Algorithm 2 Step 1: Generate sample from the Birnbaum-Saunders distribution	
Step 2: At the b step	
(a) Generate x∗=(x1∗,x2∗,...,xn∗) with replacement from x=(x1,x2,...,xn)	
(b) Compute b^(α^,α) using Eq. (7) and compute b^(β^,β) using Eq. (8)	
(c) Compute αk~ using Eq. (9) and βk~ using Eq. (10)	
(d) Compute θ^k using Eq. (11)	
Step 3: Repeat step 2, a total B times and obtain an array of θ^k’s	
Step 4: Compute Lθ.B=θ^k(γ/2) and Uθ.B=θ^k(1−γ/2)	

Bayesian approach

Bayesian approach offers a structured manner to integrate prior knowledge and revise beliefs as fresh data emerges. Assume that υ follows an inverse gamma distribution with parameters a and b, represented as IG(υ|a,b). Wang, Sun & Park (2016) utilized inverse gamma distributions as prior distributions of β and α2, denoted as IG(β|a1,b1) and IG(α2|a2,b2), respectively. The marginal distribution of β is given by

(13) p(β|x)∝β−(n+a1+1)exp(−b1β)∏i=1n((βxi)1/2+(βxi)3/2)                                        ×(∑i=1n12(xiβ+βxi−2)+b2)−n+12−a2.

The conditional posterior distribution of α2 given β is defined by

(14) p(α2|x,β)∝IG(n2+a2,12∑i=1n(xiβ+βxi−2)+b2).

The samples in Eqs. (13) and (14) are derived using Markov Chain Monte Carlo methods.

Wang, Sun & Park (2016) employed the generalized ratio-of-uniforms method to generate posterior samples for β. Wakefield, Gelfand & Smith (1991) introduced the generalized ratio-of-uniforms method. It involves a pair of random variables (u,v), each of which follows a uniform distribution as defined by

(15) A(r)={(u,v):0<u≤[p(vur|x)]1/(r+1)},r≥0

where r is a constant and p(⋅|x) is defined by using Eq. (13). Therefore, β=vur has density p(β|x)∫p(β|x)dβ. To sample random data points in A(r), the random variables (u,v) are generated from a uniform distribution over a one-dimensional bounded rectangle [0,a(r)]×[b−(r),b+(r)].

Suppose that a(r), b−(r), and b+(r) are given by

(16) a(r)=supβ>0⁡{[p(β|x)]1/(r+1)},

(17) b−(r)=infβ>0⁡{β[p(β|x)]r/(r+1)},

and

(18) b+(r)=supβ>0⁡{β[p(β|x)]r/(r+1)}.

Wang, Sun & Park (2016) proposed that a(r) and b+(r) are finite and b−(r)=0. The prospective variate β=vur is accepted if u≤[p(β|x)]1/(r+1); otherwise, the process is reiterated. The posterior samples for α2 are acquired using the LearnBayes package within the R software suite. Hence, the square root of α2 represents the posterior samples of α.

Therefore, the posterior distribution of θ is obtained as

(19) θBaye=β4(αzp+α2zp2+4)2,

where zp=Φ−1(p) is the standard normal p-th quantile.

Therefore, the 100(1−γ)% two-sided credible interval for the percentile of Birnbaum-Saunders distribution using the Bayesian approach is given by

(20) CIθ.Baye=[Lθ.Baye,Uθ.Baye]=[θBaye(γ/2),θBaye(1−γ/2)],

where θBaye(γ/2) and θBaye(1−γ/2) denote the 100(γ/2)-th and 100(1−γ/2)-th percentiles of θBaye, respectively.

Algorithm 3 was used to construct the Bayesian credible interval for the percentile of Birnbaum-Saunders distribution.

Algorithm 3 Step 1: Specify the values of a1, a2, b1, b2, and r, then compute a(r) using Eq. (16) and compute b+(r) using Eq. (18)	
Step 2: At the i step	
(a) Generate u from uniform distribution with parameters 0 and a(r), denoted as U(0,a(r))	
(b) Generate v from uniform distribution with parameters 0 and b+(r), denoted as U(0,b+(r))	
(c) Compute ρ=vur	
(d) If the value of ρ is accepted, the set β(i)=ρ if u≤[p(β|x)]1/(r+1); otherwise, repeat step (a)–step (c)	
(e) Generate λ from inverse gamma distribution with parameters n2+a2 and 12∑j=1n(xjβ(i)+β(i)xj−2)+b2, denoted as IG(n2+a2,12∑j=1n(xjβ(i)+β(i)xj−2)+b2) and compute α(i)=λ	
Step 3: Compute the posterior distribution of θ, denoted as θBaye, using Eq. (19)	
Step 4: Repeat step 2 and step 3, a total M times and obtain an array of θBaye’s	
Step 5: Compute Lθ.Baye=θBaye(γ/2) and Uθ.Baye=θBaye(1−γ/2)	

Highest posterior density approach

The HPD interval was constructed using the posterior distribution of θ as specified in Eq. (19). This interval has the smallest length among all intervals that contain 100(1−γ)% within the posterior probability. Every point within the interval has a greater probability than any point located outside of it, as explained by Box & Tiao (2011).

Therefore, the 100(1−γ)% two-sided credible interval for the percentile of Birnbaum-Saunders distribution using the HPD approach is given by

(21) CIθ.HPD=[Lθ.HPD,Uθ.HPD],

where Lθ.HPD and Uθ.HPD are determined using the hdi function within the HDInterval package of the R software suite.

Algorithm 4 was used to construct the HPD interval for the percentile of the Birnbaum-Saunders distribution.

Algorithm 4 Step 1: Specify the values of a1, a2, b1, b2, and r, then compute a(r) using Eq. (16) and compute b+(r) using Eq. (18)	
Step 2: At the i step	
(a) Generate u from uniform distribution with parameters 0 and a(r), denoted as U(0,a(r))	
(b) Generate v from uniform distribution with parameters 0 and b+(r), denoted as U(0,b+(r))	
(c) Compute ρ=vur	
(d) If the value of ρ is accepted, the set β(i)=ρ if u≤[p(β|x)]1/(r+1); otherwise, repeat step (a)–step (c)	
(e) Generate λ from inverse gamma distribution with parameters n2+a2 and 12∑j=1n(xjβ(i)+β(i)xj−2)+b2, denoted as IG(n2+a2,12∑j=1n(xjβ(i)+β(i)xj−2)+b2) and compute α(i)=λ	
Step 3: Compute the posterior distribution of θ, denoted as θBaye, using Eq. (19)	
Step 4: Repeat step 2 and step 3, a total M times and obtain an array of θBaye’s	
Step 5: Compute Lθ.HPD and Uθ.HPD	

Results

In this study, confidence intervals are proposed using the GCI approach, the bootstrap approach, the Bayesian approach, and the HPD approach. A simulation study was conducted to evaluate the performance of these confidence intervals. A Monte Carlo simulation study was carried out to evaluate the effectiveness of the suggested confidence intervals for percentile of the Birnbaum-Saunders distribution, utilizing the R software. The evaluation involved comparing the performance of these confidence intervals in terms of coverage probabilities and average lengths. The most desirable confidence interval is defined as one that achieves a coverage probability of the nominal confidence level 0.95 or higher, and the shortest average length.

According to Puggard, Niwitpong & Niwitpong (2022), normal random variables were used to generate the Birnbaum-Saunders random variables. For percentiles, the shape parameter was set as α= 0.10, 0.25, 0.50, 0.75, and 1.00, while the scale parameter was fixed at β= 1.00. Moreover, for Bayesian credible interval and HPD interval, we considered r= 2.00 and set the hyperparameter a1, a2, b1 and b2 to 10−4. We conducted 5,000 replications with 5,000 for the GCI using GPQ, B = 500 for the bootstrap confidence interval, and M = 1,000 for the Bayesian credible interval and HPD interval.

Algorithm 5 was used to compute the coverage probabilities and average lengths of the proposed confidence intervals for the percentile of Birnbaum-Saunders distribution.

Algorithm 5 Step 1: Use Algorithm 1–Algorithm 4 to construct the confidence intervals	
Step 2: If Lθ≤θ≤Uθ, set p= 1; else set p= 0	
Step 3: Compute Uθ−Lθ	
Step 4: Repeat step 1–step 3, a total 5,000 times	
Step 5: Compute mean of p defined by the coverage probability	
Step 6: Compute mean of Uθ−Lθ defined by the average length	

The findings are based on the following simulation work. The performances of the proposed confidence intervals for the percentile of the Birnbaum-Saunders distribution were presented in Table 1 and displayed in Figs. 1 and 2. From Table 1, for n≤50, the results showed that the coverage probabilities of all the proposed confidence intervals were lower than the nominal confidence level of 0.95. However, the GCI had coverage probabilities close to the nominal confidence level of 0.95. For n=100, the coverage probabilities of both the GCI and Bayesian credible interval exceeded the nominal confidence level of 0.95 in some cases. Figures 1 and 2 present the coverage probabilities and average lengths of the confidence intervals for the percentile, corresponding to various sample sizes and shape parameters, respectively. According to Fig. 1, it can be observed that the coverage probabilities were close to the nominal confidence level of 0.95 as the sample size increased. Furthermore, the average lengths of all approaches decreased as the sample size increased. Based on the simulation results presented in Fig. 2, the coverage probabilities were close to the nominal confidence level of 0.95 as the shape parameter increased. Additionally, the average lengths of all approaches increased with the shape parameter.

Table 1 The coverage probabilities and average lengths of 95% two-sided confidence intervals for the percentile of Birnbaum-Saunders distribution.

n	p	α	Coverage probability (Average length)	
CIθ.GCI	CIθ.B	CIθ.Baye	CIθ.HPD	
10	0.50	0.10	0.9494 (0.1388)	0.9008 (0.1130)	0.9354 (0.1294)	0.9320 (0.1281)	
		0.25	0.9514 (0.3503)	0.8952 (0.2842)	0.9356 (0.3261)	0.9284 (0.3217)	
		0.50	0.9490 (0.7066)	0.9020 (0.5630)	0.9334 (0.6552)	0.9330 (0.6386)	
		0.75	0.9474 (1.0770)	0.9026 (0.8353)	0.9320 (0.9837)	0.9290 (0.9434)	
		1.00	0.9484 (1.4662)	0.9044 (1.0969)	0.9340 (1.3028)	0.9352 (1.2265)	
30	0.50	0.10	0.9498 (0.0738)	0.9298 (0.0689)	0.9424 (0.0721)	0.9412 (0.0715)	
		0.25	0.9480 (0.1840)	0.9296 (0.1718)	0.9428 (0.1801)	0.9382 (0.1784)	
		0.50	0.9510 (0.3628)	0.9342 (0.3369)	0.9466 (0.3543)	0.9436 (0.3498)	
		0.75	0.9462 (0.5350)	0.9280 (0.4917)	0.9396 (0.5177)	0.9358 (0.5088)	
		1.00	0.9526 (0.6948)	0.9408 (0.6287)	0.9496 (0.6623)	0.9466 (0.6475)	
50	0.50	0.10	0.9422 (0.0564)	0.9308 (0.0540)	0.9400 (0.0556)	0.9352 (0.0552)	
		0.25	0.9488 (0.1408)	0.9372 (0.1349)	0.9446 (0.1388)	0.9436 (0.1376)	
		0.50	0.9490 (0.2754)	0.9398 (0.2625)	0.9460 (0.2711)	0.9438 (0.2681)	
		0.75	0.9490 (0.4035)	0.9356 (0.3812)	0.9446 (0.3937)	0.9426 (0.3884)	
		1.00	0.9512 (0.5232)	0.9382 (0.4869)	0.9444 (0.5027)	0.9432 (0.4946)	
100	0.50	0.10	0.9560 (0.0396)	0.9466 (0.0385)	0.9506 (0.0392)	0.9486 (0.0389)	
		0.25	0.9472 (0.0982)	0.9374 (0.0957)	0.9452 (0.0974)	0.9414 (0.0966)	
		0.50	0.9526 (0.1925)	0.9438 (0.1869)	0.9490 (0.1903)	0.9462 (0.1885)	
		0.75	0.9486 (0.2804)	0.9442 (0.2704)	0.9500 (0.2754)	0.9476 (0.2724)	
		1.00	0.9520 (0.3595)	0.9428 (0.3419)	0.9494 (0.3481)	0.9454 (0.3440)	
Note:

Bold font means the confidence interval with coverage probability greater than or equal to 0.95 and the shortest average length.

Figure 1 Comparison of the coverage probabilities and average lengths of the confidence intervals for the percentile according to sample sizes.

(A) Coverage probability (B) Average lengths.

Figure 2 Comparison of the coverage probabilities and average lengths of the confidence intervals for the percentile according to shape parameters.

(A) Coverage probability (B) Average lengths.

Empirical application

The GCI, bootstrap, Bayesian, and HPD approaches can be employed to calculate confidence intervals for the percentile of PM2.5 levels in Mae Hong Son province and Lampang province, Thailand.

Table 2 contains data on daily PM2.5 levels reported by the Pollution Control Department from January 1 to June 30, 2023. Figure 3 provides histograms illustrating the distribution of daily PM2.5 levels in Mae Hong Son and Lampang provinces. Due to the positively skewed nature of the data, seven distribution models were considered: normal, log-normal, Weibull, gamma, exponential, Cauchy, and Birnbaum-Saunders distributions. The suitability of these models for fitting the daily PM2.5 level data was assessed using the Akaike Information Criterion (AIC). Table 3 displays the AIC values for these seven probability models, calculated based on the PM2.5 level data from both provinces. The results in Table 3 indicated that the Birnbaum-Saunders distribution is the most appropriate model for fitting the daily PM2.5 level data in both Mae Hong Son and Lampang provinces, as it yields the lowest AIC value.

Table 2 Daily PM2.5 levels data in Mae Hong Son and Lampang provinces.

Provinces	Daily PM2.5 levels ( μg/m3)	
Mae Hong Son	25	21	19	18	18	20	16	23	25	26	
	20	18	17	21	27	28	24	28	27	28	
	25	21	26	30	36	38	40	42	41	35	
	40	41	47	42	41	34	30	31	37	39	
	54	64	78	69	57	70	68	13	9	15	
	21	27	28	34	36	48	59	66	75	78	
	79	92	97	103	104	91	91	111	118	142	
	216	134	53	59	62	60	98	116	104	148	
	137	144	255	309	244	237	209	263	321	256	
	233	144	109	147	176	204	203	147	117	108	
	124	119	104	106	166	214	138	61	59	56	
	53	50	38	34	34	49	39	36	50	49	
	34	23	23	22	28	29	31	26	13	18	
	8	8	5	11	12	13	17	18	19	19	
	21	19	19	15	17	17	16	17	13	10	
	11	8.3	11.6	13.3	10.3	6.2	5.5	5.8	7.1	14.1	
	13.5	15.2	10	10.1	10.2	10.2	9	5.5	3.7	3.7	
	3.9	3.6	3.3	4.2	4.2	4.4	4.4	5	5.5	4.8	
	7.1										
Lampang	29	30	28	30	27	21	19	33	39	32	
	37	33	30	27	32	41	43	39	41	40	
	44	50	49	57	60	59	71	62	59	67	
	88	87	80	82	68	52	40	39	65	106	
	108	113	125	158	211	153	42	23	19	27	
	57	67	74	72	88	92	59	96	143	155	
	150	138	132	119	100	84	90	97	101	99	
	103	64	49	48	53	41	51	34	32	39	
	42	51	62	129	119	99	121	94	103	148	
	148	86	64	92	172	185	182	113	80	72	
	96	133	130	100	122	119	82	89	91	93	
	91	60	55	37	61	48	30	38	44	34	
	36	63	32	37	48	52	51	46	17	19	
	21	20	16	17	22	23	23	35	33	31	
	36	27	24	23	29	31	28	34	25	22	
	18	18.6	17.4	20.3	20.6	16.1	16.8	16.7	14.9	16.5	
	19.2	20.4	21.6	18.7	16.91	8.2	16.3	16	14.8	15.6	
	14.8	13.5	13.3	14.7	12.8	13	12.5	12.7	14.4	13	
	12.7										
Note:

Pollution Control Department, Thailand http://air4thai.pcd.go.th/webV3/#/History.

Figure 3 Histograms of the daily PM2.5 level data for (A) Mae Hong Son and (B) Lampang provinces.

Table 3 The estimated AIC values for the probability models using the PM2.5 level data from Mae Hong Son and Lampang provinces.

Distributions	Mae Hong Son province	Lampang province	
Normal	2,031.7990	1,876.4660	
Log-normal	1,818.1560	1,787.6230	
Weibull	1,836.4100	1,802.0070	
Gamma	1,836.9690	1,795.2050	
Exponential	1,834.9720	1,836.3210	
Cauchy	1,946.9400	1,910.7870	
Birnbaum-Saunders	1,812.9130	1,782.0890	
Note:

Bold font means the distribution with the lowest AIC value.

Table 4 reports the statistics of daily PM2.5 level data in Mae Hong Son and Lampang provinces. Table 5 displays 95% two-sided confidence intervals for the percentile of daily PM2.5 level data in these provinces, using the GCI, bootstrap, Bayesian, and HPD approaches. The results reveal that all confidence intervals encompass the true percentiles. Regarding interval length, the HPD approach yielded the shortest results for daily PM2.5 level data percentiles, while the GCI approach produced the longest results. However, in simulation, the lengths of the bootstrap, Bayesian, and HPD approaches were shorter than the GCI approach, but their coverage probabilities were below the nominal confidence level of 0.95. Additionally, the coverage probability and average length in the simulation were calculated using 5,000 random samples, whereas the length in the example was computed using a single sample. Consequently, the bootstrap, Bayesian, and HPD approaches are not recommended for constructing confidence interval for percentile.

Table 4 Sample statistics for the daily PM2.5 level data in Mae Hong Son and Lampang provinces.

Statistics	Mae Hong Son province	Lampang province	
Minimum	3.30	8.20	
Mean	58.18	58.39	
Maximum	321.00	211.00	
n	181	181	
α^	1.25	0.78	
β^	31.93	45.46	
θ^	31.93	45.46	

Table 5 The lower limit ( Lθ) and upper limit ( Uθ) of the 95% confidence intervals for the percentile of the daily PM2.5 level data in Mae Hong Son and Lampang provinces.

Approaches	Mae Hong Son province	Lampang province	
[Lθ,Uθ]	Interval length	[Lθ,Uθ]	Interval length	
GCI	[27.8787,38.3599]	10.4812	[40.0647,49.7619]	9.6972	
Bootstrap	[28.0199,38.1855]	10.1656	[40.0333,49.6188]	9.5855	
Bayesian	[27.9843,37.7447]	9.7604	[40.5136,49.7726]	9.2590	
HPD	[28.0339,37.7767]	9.7428	[40.4549,49.5468]	9.0919	

Discussion

In the field of environmental sciences and air quality, percentiles serve as a means to characterize the majority of PM2.5 levels. Utilizing the confidence interval for the percentile of PM2.5 levels enables an estimation of the predominant PM2.5 levels. As a result, leveraging the estimated percentile of PM2.5 levels can contribute to strategic planning for the reduction of air pollutants.

Puggard, Niwitpong & Niwitpong (2022) introduced the HPD approach for constructing confidence intervals for the variance and difference of variances of Birnbaum-Saunders distributions. Nevertheless, in certain situations, percentiles may be more suitable than variance. Hence, the aim of this study was to estimate the percentiles of the Birnbaum-Saunders distribution. Confidence intervals for the percentile of Birnbaum-Saunders distribution were generated using the GCI, bootstrap, Bayesian, and HPD approaches. The GCI approach exhibited strong performance in constructing these intervals. All four approaches utilized simulation data to create these confidence intervals. The GCI approach leverages GPQs for interval construction, while the bootstrap approach relies on the sampling distribution. In contrast, the Bayesian and HPD approaches are rooted in prior distributions. The study results suggest that the GCI approach is the preferred method for constructing confidence intervals for the percentile of the Birnbaum-Saunders distribution. This conclusion is consistent with the findings of previous studies by Ye, Ma & Wang (2010), Thangjai, Niwitpong & Niwitpong (2018), and Thangjai & Niwitpong (2022).

Conclusion

The confidence intervals for the percentile of the Birnbaum-Saunders distribution were established using four approaches: GCI, bootstrap, Bayesian, and HPD approaches. According to simulation results, the average lengths of the bootstrap, Bayesian, and HPD approaches were shorter than the GCI approach, but their coverage probabilities fell below the nominal confidence level of 0.95. As a result, the bootstrap, Bayesian, and HPD approaches are not recommended for constructing confidence interval for percentile. The findings consistently highlight the GCI approach as the most reliable in terms of coverage probability. Therefore, the GCI approach is recommended for constructing confidence interval for percentile.

Supplemental Information

Supplemental Information 1 Daily PM2.5 levels data in Mae Hong Son and Lampang provinces.

Source: Pollution Control Department, Thailand. http://air4thai.pcd.go.th/webV3/#/History.

Supplemental Information 2 R code to construct histograms of PM2.5 data from (A) Mae Hong Son Province (B) Lampang Province.

Supplemental Information 3 R code to construct confidence intervals from all methods in this article.

Additional Information and Declarations

Competing Interests

Author Contributions

Data Availability

The authors declare that they have no competing interests.

Warisa Thangjai conceived and designed the experiments, performed the experiments, prepared figures and/or tables, authored or reviewed drafts of the article, and approved the final draft.

Sa-Aat Niwitpong analyzed the data, authored or reviewed drafts of the article, and approved the final draft.

Suparat Niwitpong conceived and designed the experiments, analyzed the data, prepared figures and/or tables, authored or reviewed drafts of the article, and approved the final draft.

The following information was supplied regarding data availability:

The data and R code for computing coverage probability and average length of all confidence intervals is available in the Supplemental Files.

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
