# Peer review of "Estimation of the percentile of Birnbaum-Saunders distribution and its application to PM2.5 in Northern Thailand"

_PeerJ, doi:10.7717/peerj.17019_

## Round 0.1 · original submission · Major Revisions

Dear Authors,

Considering the feedback from both reports and the analysis, the best decision regarding the manuscript would be to recommend a major revision. Here are the key reasons:

- Common recommendations of the peer-reviewers: Both reports highlight the need for improvements in referencing and providing a literature review. These are essential aspects that can enhance the overall quality of the manuscript.

- Specific Suggestions from Report 1: Report 1 identifies several areas that require attention, such as revising the experimental design and validity of findings. Specific issues in different sections (Introduction, Material and Methods, Results, Discussion, and Conclusion) need to be addressed to enhance clarity, organization, and completeness.

- Specific Suggestions from Report 2: Some revisions are needed, particularly in briefly mentioning methods based on Puggard et al. (2022) without detailing them and emphasizing the innovation and value of the article.

Overall comments and recommendations from editor:

The reviewers have not raised concerns about the fundamental validity of the findings, but they emphasize the importance of addressing specific issues related to clarity, completeness, and referencing.

The authors are invited to:

- revise the introduction and discussion by incorporating additional references and a literature review on PM2.5.

- carefully describe the study domain, duration, and the number of stations in the Material and Methods section.

- correct the order of results, clarify the mention of the simulation study, and move some formulas to supplementary material.

- provide a more detailed discussion on the value of findings, comparing them with similar studies.

- expand and improve the conclusion section.

Thank you!

·

Basic reporting

More references are required to be added to the introduction and discussion sections.

Experimental design

It needs some revision.

Validity of the findings

It needs some revision.

Additional comments

Introduction
Lines 27-47 : please add references and also do a literature review on PM2.5 as a air pollutant. You can add below mentioned papers to your introduction and reference list:

https://doi.org/10.1016/j.jes.2023.11.019
https://doi.org/10.3390/su13042201

Material and methods:
You have not mentioned the study domain, the duration of study, the number of stations, so please describe you study area and data collection carefully.
You can move some of formulas to supplementary material, it is confusing in the main text.
Results:
It is confusing, the order of results is not correct. You need reorder it.
Line 311: what simulation study? You have not mentioned it!
Discussion:
You have not mentioned that why your findings can be valuable in environmental sciences and air quality, please discuss it in more detail.
You have not compared your results with similar studies, so please compare it to show the importance of your job or possible extra findings compared to other studies.
Conclusion:
Improve it, it is a very small conclusion. Talk about the methods you have used and your findings and also the study area. Please re write it.

Reviewer 2 ·

Basic reporting

In the paper, authors consider the problem for confidence intervals of the percentile of Birnbaum-Saunders distribution, which are based on the generalized confidence interval (GCI) approach, the bootstrap approach, the Bayesian approach, and the highest posterior density (HPD) approach, respectively.

The methods are not new, while are based on those developed by Puggard et al. (2022 ) as follows.
Puggard W, Niwitpong S-A, Niwitpong S. 2022. Confidence intervals for the variance and difference of388
variances of Birnbaun-Saunders distributions. Journal of Statistical Computation and Simulation 92(13): 389 2829-2845 DOI 10.1080/00949655.2022.2050231.

The methods in the two papers are completely consistent, except that this paper deals with quantiles, while the by Puggard et al. (2022 ) considers variance.

Therefore, the authors do not need to describe thees methods in detail again, just briefly mention it. I suggest the author should address the innovation and value of the article.

Experimental design

Methods should bebe accurately described and cited.

Validity of the findings

No comment.

Additional comments

In the paper, authors consider the problem for confidence intervals of the percentile of Birnbaum-Saunders distribution, which are based on the generalized confidence interval (GCI) approach, the bootstrap approach, the Bayesian approach, and the highest posterior density (HPD) approach, respectively.

The methods are not new, while are based on those developed by Puggard et al. (2022 ) as follows.
Puggard W, Niwitpong S-A, Niwitpong S. 2022. Confidence intervals for the variance and difference of388
variances of Birnbaun-Saunders distributions. Journal of Statistical Computation and Simulation 92(13): 389 2829-2845 DOI 10.1080/00949655.2022.2050231.

The methods in the two papers are completely consistent, except that this paper deals with quantiles, while the by Puggard et al. (2022 ) considers variance.

Therefore, the authors do not need to describe thees methods in detail again, just briefly mention it. I suggest the author should address the innovation and value of the article.

---

## Round 0.2 · accepted · Accept

Dear Authors,

I am pleased to inform you that your manuscript titled "Estimation of the percentile of Birnbaum-Saunders distribution and its application to PM2.5 in Northern Thailand " has undergone thorough review, and I am delighted to confirm that all of the reviewers' comments have been satisfactorily addressed.

Furthermore, based on the revisions made in response to the reviewers' comments and the overall quality of the manuscript, I am pleased to state that your manuscript is now considered ready for publication.

I would like to extend my congratulations to all co-authors on this achievement. Your dedication to addressing the reviewers' feedback and the quality of your work are commendable.

Thank you for your hard work and diligence throughout the review process. I look forward to seeing your manuscript published in PeerJ.

·

Basic reporting

no comment

Experimental design

no comment

Validity of the findings

no comment

Reviewer 2 ·

Basic reporting

no comment

Experimental design

no comment

Validity of the findings

no comment

Additional comments

no comment